# Normal faulting and viscous buckling in the Tibetan Plateau induced by a weak lower crust

Sarah H. Bischoff [1] & Lucy M. Flesch [1]

Flow of weak lower crust has been invoked to reconcile observed topographic gradients, uniform elevations, slow seismic velocity, and high conductivity measured in the Tibetan Plateau, with viscosity estimates of $10^{16}$–$10^{21}$ Pa·s. Here we investigate the dynamic response resulting from a range of lower crust viscosities in a 3-D lithospheric-scale geodynamic simulation of the India–Eurasia collision zone to determine bounds of physically viable lower crustal strengths. We show that thickening of the plateau is accommodated through viscous buckling of the upper crust in response to lower crustal flow for a lower crustal viscosity on the order of $10^{20}$ Pa·s. This generates two east–west trending bands of surface subsidence and dilatation consistent with observed normal faulting and estimates of vertical velocity. These results suggest viscous buckling of the upper crust, induced by lower crustal flow from gravitational pressure gradients due to high topography, is responsible for the observed extension in Tibet.

[1] Department of Earth, Atmospheric & Planetary Sciences, Purdue University, West Lafayette, IN 47907, USA. Correspondence and requests for materials should be addressed to L.M.F. (email: lmflesch@purdue.edu)

 

The Tibetan Plateau, with an average elevation of 5 km over a lateral scale of 1000s of km, exhibits a complex pattern of deformation. It is well established the large-scale north–south compression and lithospheric thickening results from collision of the continental Indian and Eurasian plates; however, questions remain as to the mechanism responsible for generating the observed east–west extension, normal faulting, and subsidence extending parallel to the margin. Proposed hypotheses include gravitational collapse of high topography, convective removal of a thickened litho-spheric mantle, reduction in rate of Indian convergence, spreading of weak Eurasian crust over subducted Indian slab, interaction between Eurasian crust and discontinuities in sub-ducted Indian lower crust, and geometry of Indian collision[1–8]. The dynamics of the Tibetan Plateau have been widely and successfully approximated by thin viscous sheet (TVS) models, which assume negligible horizontal shear and treat the litho-sphere as a homogeneous sheet with only lateral strength variations[1,2] (and references therein). However, observations of high conductivity[9], slow seismic velocity[10], and radial aniso-tropy[11] have been interpreted as a pervasive weakness in the Tibetan lower crust that could indicate deviation from the TVS assumptions[12]. In addition, flow of weak lower crust has been posited to reconcile crustal thickening of the eastern plateau in the absence of appreciable upper crustal shortening, variable topographic gradients from high-to-low elevations[13], and low plateau surface relief[14] with viscosity estimates ranging from $10^{16}$ to $10^{21}$ Pa·s[13,15,16]. Laboratory-derived lithospheric strength envelopes[17] predict low strength in continental lower crust, presumably augmented by anomalously high crustal temperatures in Tibet[18] and perhaps associated with enhanced radioactive heat production in the doubly-thick crust[19].

Previous geodynamic simulations of lower crustal deformation in Tibet have primarily focused on estimating the viscosity required to generate observed topographic relief[15] and gradients[13] along 2D profiles, or full 3-D time-dependent thermomechanical derivation of viscosity with assumed flow laws and temperature gradients[20–22] to assess the effects of crustal thickness and vis-cosity variations and required driving forces in generating both topographic features and continental subduction[23–25]. Since 2D simulations neglect flow in and out of the third dimension and thermomechanical simulations derive strength distributions rather than test a given hypothetical distribution determined from geophysical data, neither approach addresses the lithospheric wide influence of an assumed lower crustal strength.

In this work we perform 3-D lithospheric-scale simulations of the India–Eurasia (IN-EU) collision zone, varying lower crustal strength for published viscosity estimates ranging from $10^{19}$ to $10^{21}$ Pa·s to explicitly determine the lithospheric surface response and assess the level of lower crustal flow, for an assumed layer-averaged lower crustal strength. Geodynamic simulations are governed by incompressible steady state Stokes-flow within a 100 km thick spherical cap simulation driven by edge velocity conditions and gravity acting on 3-D varying material properties. We use a viscous rheology to estimate the accumulation of stress and deformation over multiple seismic cycles, and approximates the depth-varying material property of the upper crust. We divide the model geometry between Indian plate indenter and Eurasian upper crust, lower crust, and lithospheric mantle (see Fig. 1a, b and Methods), assuming lateral bounds of weak lower

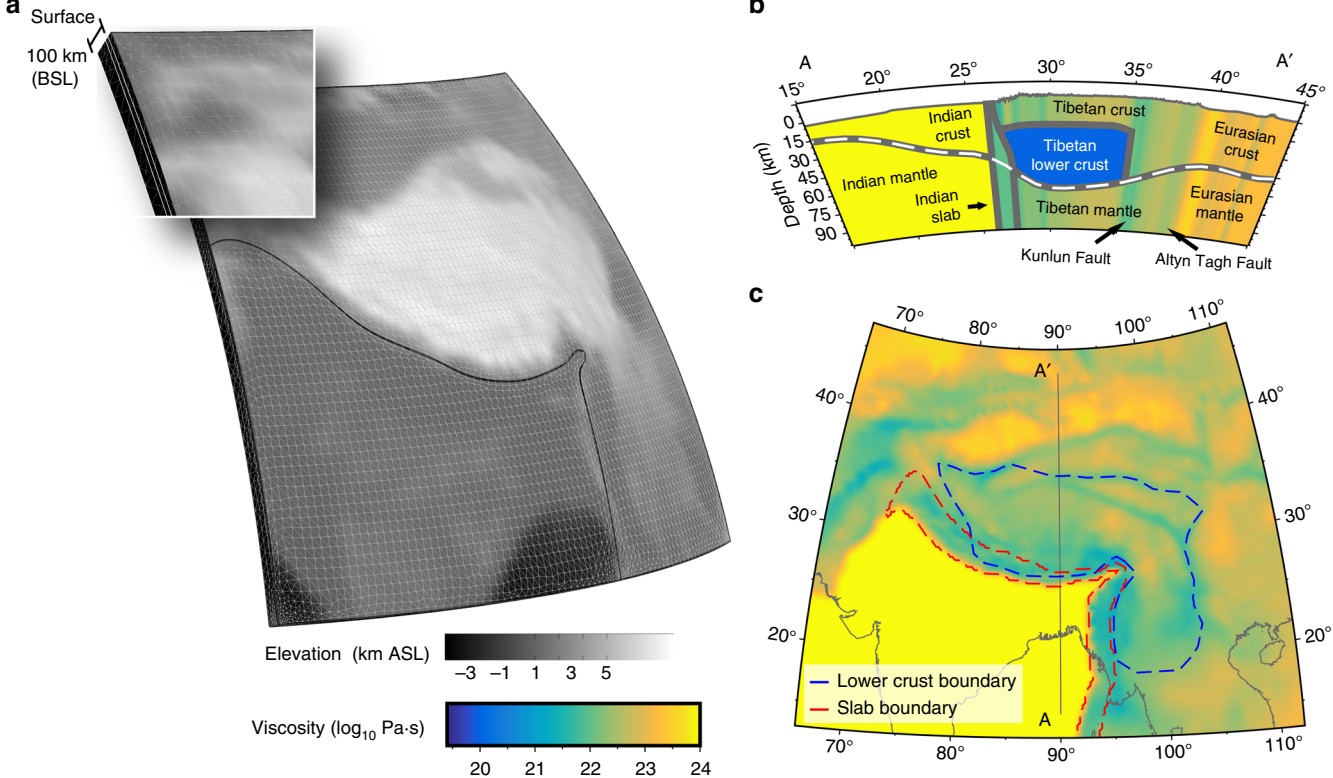

**Fig. 1** Schematic of model geometry and viscosity distribution. **a** Geometry with color scale representing elevation above sea level, gray lines delineating mesh elements, and black line showing Indian/Asian domain boundary. Inset shows enlarged geometry corner, with white lines showing Asian upper/lower crust and mantle domains. **b** Cross-section of model at 90° E with color scale representing viscosity variation, gray lines delineating regions with vertically coherent viscosity, and dashed white line showing Moho. **c** Map-view of model showing vertically-averaged lithospheric effective viscosity and dashed blue/red lines outlining surface projections of weak lower crust/slab regions. A–A′ line locates cross-section in **b**

crust in Tibet correspond to zones of slow seismic velocity[10,26,27], high conductivity[9,28], and aseismicity[27,29] (Fig. 1c). Becaus several studies have demonstrated the importance of lateral strength heterogeneity and pre-existing lithospheric structure in generating geophysical features that correlate with the Tibetan Plateau[1,20,23,24,30], we use the lateral strength heterogeneity based on the updated vertically-averaged lithospheric effective viscosity estimates of Flesch et al.[1,30] (Fig. 1). This geophysical observation based laterally varying effective viscosity field has a strong Indian plate, Tarim Basin, Gobi platform, and Sichuan Basin. Overall, Tibet, the Pamir, and Tien Shan are two orders of magnitude weaker than the stronger blocks. Additionally, even weaker regions are estimated in areas of well-developed faults (southern Tibet, Altyn-Tagh Fault, Kunlun Fault, Chaman Fault, Xianshuihe Fault, Jiali Fault, Saigan Fault, and the Himalayan Front). In order to determine a 3-D viscosity structure we require that where a weaker lower crust is present (dashed blue lines Fig. 1c) the reductions in the lower crustal layer viscosity imply increased upper crust/lithospheric mantle strength given a one-to-one ratio between upper crust and mantle strength[31] including a high viscosity zone beneath Tibet representing the subducted Indian slab[32] (dashed red lines Fig. 1c). Here we focus on isolating the surface deformation response to vertical strength heterogeneity produced by the inclusion of a weak lower crust, vary lower crustal viscosities and utilize a constant 3-D density distribution estimated from CRUST 1.0[33] for all model simulations. We find Poiseuille flow of a weak lower crust, on the order of $10^{20}$ Pa·s, induced by gradients of gravitational potential energy resulting from high topography causes the strong upper crust to viscously buckle generating bands of east–west uplift, surface subsidence and dilatation that drives normal faulting in southern and central Tibet.

## Results

### Impact of lower crustal strength on mechanism of deformation.
All three simulations for lower crustal viscosities ranging from $10^{19}$ to $10^{21}$ Pa·s, as well as a simulation with uniform crustal strength (TVS), produce similar horizontal surface velocities (Fig. 2a), indicating that the strength variations at depth are indistinguishable when using horizontal surface motions alone[30]. However, the predicted surface vertical velocity for each of the solutions demonstrates a strong dependence on lower crustal strength (Fig. 3). The weak lower crust, squeezed between the converging Indian plate to the south and strong lithospheric blocks to the north and east[20], Tarim and Sichuan basins respectively, deforms under ductile simple shear in simulations when the lower crustal viscosity is $10^{21}$ Pa·s or larger, producing pronounced uplift along the Himalayan front and distributed uplift across Tibet (Fig. 3e, f). In contrast, for lower crustal viscosities of $10^{20}$ Pa·s and below, Poiseuille flow begins to develop wherever lateral crustal thickness variations generate the requisite gravitational pressure gradients due to the thick crust and high topography (Fig. 3k, l). Additionally in simulations developing Poiseuille flow, the large strength contrast between upper and lower crust prevents deformation in the stronger upper crust from occurring at the same rate as the underlying fast-flowing, weak lower crust, causing the upper crust to viscously buckle[34] (and references therein) to maintain plateau continuity. The viscous buckling generates alternating east–west bands of uplift and subsidence at a wavelength of ~ 320 km over the portion of the plateau underlain by weak lower crust (Fig. 3g, h). Upper-to-lower crust strength contrasts can exceed two orders of magnitude in these simulations, locally decoupling motion of the upper and lower crust[12]

Comparison between simulated and observed crustal deviatoric stresses indicate a weak lower crustal viscosity on the order of $10^{20}$ Pa·s is able to reproduce observations, in agreement with other studies[15,21,35], and produces the best fit statically to observed Global Positioning System (GPS) data for each case (Table 1). Since the division of lithospheric-averaged viscosity into layers of distinct strength is inherently a non-unique process, we cannot preclude the case of weaker lower crustal viscosities provided the layer of weakness is thinner than that we have modeled here[16]. However, assuming our inferred model geometry is appropriate for IN-EU geodynamics, only a weak lower crust viscosity of $10^{20}$ Pa·s successfully reproduces the observed pattern of compression at the Himalayas, east–west tension in southern and northern Tibet, north–south tension in eastern Tibet, and east–west tension in Yunnan (Fig. 2c). Simulations with weaker lower crustal strength produce crustal tensional deviatoric stresses at significant angles to observed normal faulting.

In order to be confident the observed viscous buckling is not an artifact of the estimated geophysical observationally based lateral strength distribution (Fig. 1), we perform two additional sets of numerical block simulations where we first assume Indian lithosphere is $10^{24}$ Pa·s, the Asian lithosphere is $10^{23}$ Pa·s and weak lower crust of $10^{20}$ Pa·s. In the second simulation, we assume the Indian lithosphere is $10^{24}$ Pa·s, the Asian lithosphere is $10^{23}$ Pa·s, the Tibetan lithosphere is $10^{22}$ Pa·s and weak lower crust of $10^{20}$ Pa·s. We find that viscous buckling occurs regardless of lateral variations in effective viscosity (Supplementary Figure 1a, b), it is only dependent on the location and extent of the weaker lower crust and the strength contrast between the upper and lower crust such that Poiseuille flow develops. However, the predicated surface horizontal velocity field and crustal deviatoric stress fields for these two simulations produce a degraded fit with observations in comparison to simulations utilizing a geophysical observationally based estimated lateral strength distribution (Table 1). These block simulations yield further support that weakening the lower crust sufficiently for the development of Poiseuille flow will lead to rates of deformation in the lower crust that are too high for the upper crust to maintain and thus requires the upper crust to viscously buckle in order to maintain plateau continuity. The extent of buckling in the upper crust is contained to regions that overlay a weaker lower crust.

Due to the fact that seismic and MT studies have argued that the weak lower crustal layer in Tibet extends to 50 km depth[9,10], we perform one final simulation to investigate the effect of thickness of the weak lower crustal layer on viscous buckling. This final simulation is identical to that presented in Fig. 3c, g, k with the exception that the weak lower crustal layer is uniform everywhere extending from 20 km to 50 m depth. We find that the presence of a weak lower crustal layer that does not extend to the Moho will still develop Poiseuille flow inducing viscous buckling in the upper crust (Supplementary Figure 1c) and produces a fit to the observed GPS data on the order of the case with a thicker weak lower crust with a viscosity of $10^{20}$ Pa·s (Table 1).

### Vertical motion as a proxy for dynamics of lower crust.
As noted by Bendick and Flesch[34], we find that horizontal surface velocities are nearly indistinguishable within uncertainty across the entire region of simulated vertical strength distributions tested here (Fig. 2a). However, we find significant variance between vertical surface motions produced in simulations with different lower crustal strengths (Fig. 3e–h), hence observations of vertical motion in Tibet provide the additional constraints in order to distinguish between physically viable numerical simulations. Paleoelevation estimates derived from fossil flora enthalpy[36] and isotope–elevation

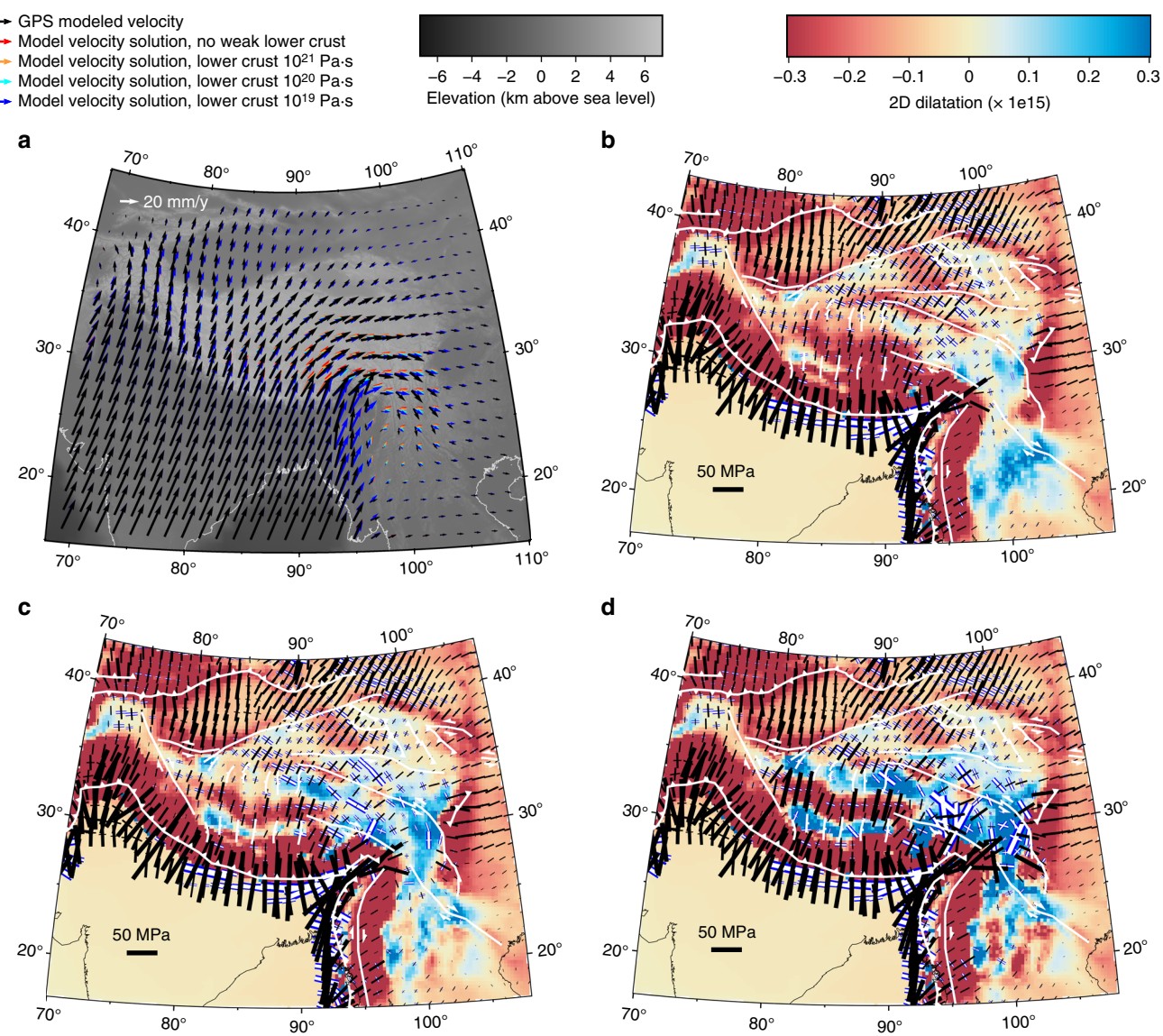

**Fig. 2** Modeled solutions of horizontal surface motion and crustal principal deviatoric stresses/dilatation rates. **a** Arrows show horizontal components of surface deformation modeled from GPS observations (black), and test cases with lower crust characterized by no weak lower crust (red), and weak lower crust viscosities of $10^{21}$ (orange), $10^{20}$ (cyan), and $10^{19}$ (navy) Pa·s on gray-scale showing topography. Crustal dilatation rates (in blue/red) and principal compressive (black bars) and extensive (white bars outlined in blue) deviatoric stresses are shown for test cases with lower crust viscosity of **b** $10^{21}$, **c** $10^{20}$, and **d** $10^{19}$ Pa·s. White lines mark major faults

relationships[37–44] across the IN-EU collision zone reveal a spatially varying pattern of uplift and subsidence since the Miocene (Fig. 4). Additionally, Liang et al.[45] approximate present-day vertical tectonic motion of Tibet by relating vertical GPS observations across Tibet to measurements from three continuous stations located on stable blocks to the north. While paleoelevation estimates provide no constraints on uplift rate and age estimates of individual sites vary widely from 45 to 5 Ma, both the paleoaltimetry and geodetic results reveal an alternating pattern of margin parallel uplift and subsidence across southern Tibet roughly coincident with the surface uplift and subsidence patterns produced by simulations with weak lower crust of $10^{20}$ Pa·s and below (Fig. 3g, h). However, simulations with lower crustal viscosity $10^{19}$ Pa·s and below produce high rates of lower crustal mass flux (in some places > 40 mm/yr), resulting in unphysical rates of uplift and subsidence (Figs 2d and 3h, l). Thus, our simulations bound lower crustal viscosity as <$10^{21}$ Pa·s and >$10^{19}$ Pa·s. Additionally, these simulations yield high rates of uplift at the eastern and western syntaxes, in

good agreement with observations of very high rates of recent rock uplift[46].

**Spatial correlation in gravity lows and simulated buckling.** Two parallel east–west trending bands of gravity lows evident from terrestrial and satellite gravity observations at wavelengths of 150–500 km have been variably interpreted as folding of the Moho in response to tectonic compression or underthrusting of the Indian lower crust[47–49]. The locations of the gravity lows are spatially well correlated with the simulated bands of surface subsidence, corresponding to the buckling of the simulated upper crust (Fig. 3g, h). Thus, gravity observations may also be sensitive to the proposed buckling in the upper crust. We posit, if the crust and the mantle are of the same strength as they are in the simulations presented here, when Poiseuille flow of the lower crust develops the lithospheric mantle would likewise buckle for plateau continuity, and because the spatial scale is controlled by

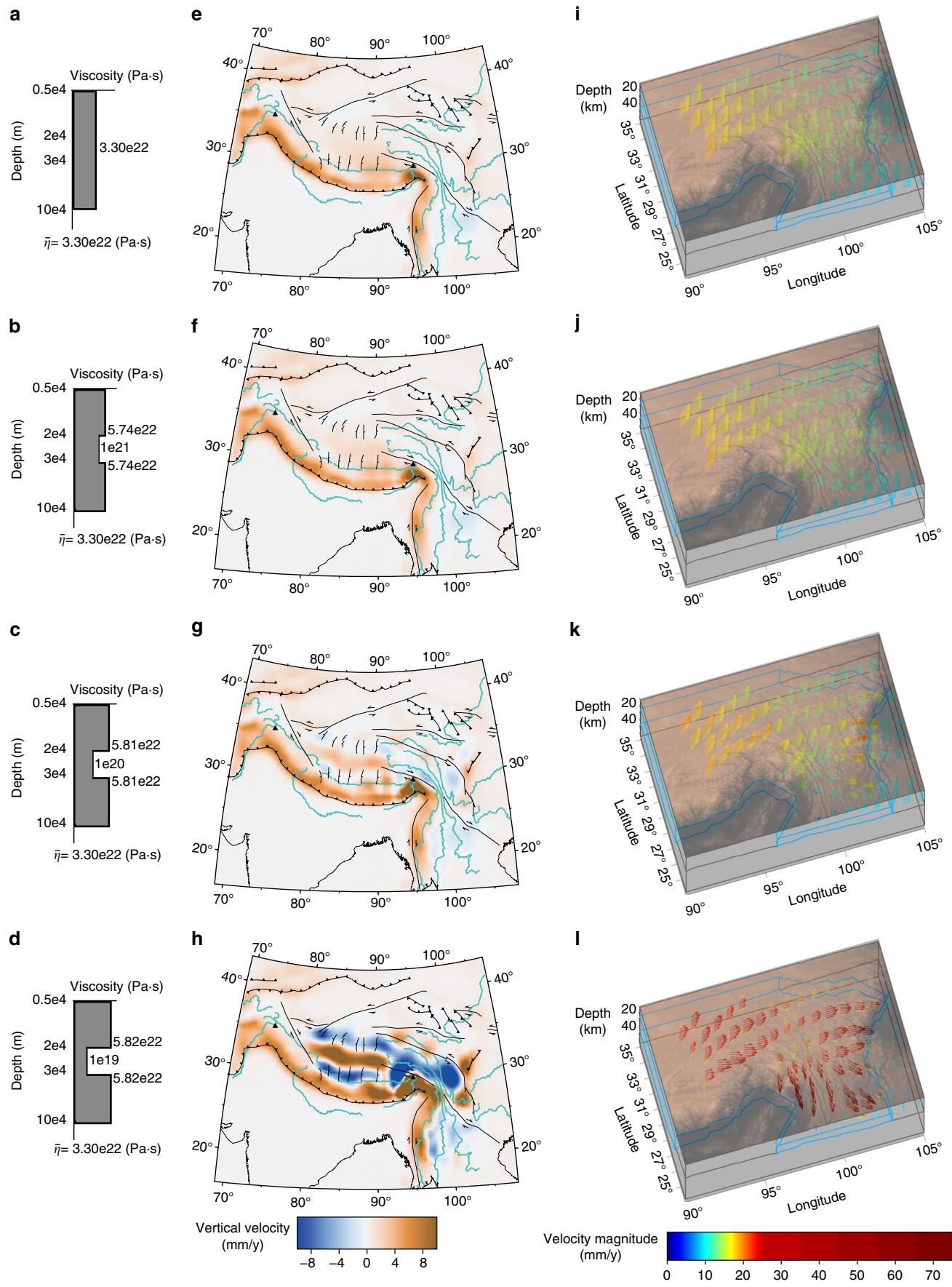

**Fig. 3** Modeled solutions of vertical surface motion and lower crust velocity. Viscosity-depth profiles at 30° N, 99° E for test cases with lower crust characterized by **a** no weak lower crust, **b** $10^{21}$, **c** $10^{20}$, and **d** $10^{19}$ Pa·s. **e–h** Model solutions with color scale representing surface vertical velocity, teal lines/black symbols marking major rivers/faults, and black triangles denoting locations of Nanga Parbat and Namche Barwa peaks. **i–l** Model velocity solutions in weak lower crust of Southeast Tibet, with arrow color representing magnitude, blue/gray lines outlining weak lower crust/other domains, and semi-transparent copper color scale showing topography

**Table 1 Misfit statistics between model prediction at GPS sites and GPS data. The latter are derived from Liang et al.[45]**

| Model | RMS$_{misfit}$ | WRMS$_{misfit}$ |
|---|---|---|
| No weak lower crust | 6.43 | 4.78 |
| $10^{21}$ Pa·s lower crust | 6.07 | 4.51 |
| $10^{20}$ Pa·s lower crust | 5.19 | 3.86 |
| $10^{19}$ Pa·s lower crust | 4.80 | 3.57 |
| Case 1-block (three viscosities) | 7.42 | 4.62 |
| Case 2-block (four viscosities) | 6.82 | 4.25 |
| Thinner weak lower crust | 5.00 | 3.09 |

the presence of weak lower crust one would expect them to correlate.

**Origins of normal faulting in southern and central Tibet.** We observe a correspondence between the simulated bands of subsidence, deviatoric stresses and patterns of dilatation for a weak lower crust on the order of $10^{20}$ Pa·s with available paleoelevation and geodetic observations of vertical motion and normal faulting in southern Tibet. In part owing to poor fieldwork conditions[45], no paleoelevation or vertical geodetic estimates have been produced for central Tibet where simulations of viscous buckling predict a second band of subsidence centered in western Qiangtang. However, we note the correlation of observed normal faulting in southern and central Tibet[5] with the pattern of subsidence, crustal thinning (Fig. 3g) and dilatation (Fig. 2c) predicted by simulations with viscous buckling. Based on these correlations, we propose that normal faulting, extension and subsidence in southern and central Tibet is a result of viscous buckling of the Tibetan upper crust in response to faster rates of deformation and flow of the weak lower crust associated with gravitational collapse. Early crustal thickening and uplift[50] of Tibet led to radioactive heating and weakening of the lower crust[19]. Growth of topography and weakening of the lower crust eventually led to the development of gravitational-induced pressure gradients and high rates of Poiseuille flow of the weak lower crust, inducing viscous buckling in the stronger upper crust and initiating faulting in the late Miocene that continues to present-day[3].

Normal faulting in Tibet has been attributed to convective removal of the Tibetan mantle[2], gravitational collapse of high topography[1], geometry of the Indian collision[8], subduction of Indian basement ridges[7], a reduction in rate of Indian convergence[4], presence of a weak lower crust[25], or spreading of weak Eurasian crust over subducted Indian slab[3]. Fast seismic velocity anomalies interpreted as Indian and Tibetan lithosphere[51] suggests removal may not have been pronounced or plateau-wide, and that a change in collision boundary condition was not widespread. Similarly, paleoelevation evidence is not conclusive on when the Tibetan Plateau reached its present-day elevation, with many studies suggesting it may have reached high elevations well before extension began in the Miocene[41], although some have argued for extension in southern Tibet starting in the early stage of the India/Eurasia collision[52,53]. Additional deformation features, including symmetrical fanning of normal fault

stress trajectories about the center of the Himalayan arc[8] and the difference in character between extension in southern and northern Tibet[3,6], can provide more constraints to identify the mechanism of extension. Armijo et al.[3] attribute the reduction in extension rate from southern to northern Tibet in terms of the greater impact of strike-slip faulting in the northern plateau. Conversely, Styron et al.[6] suggest subduction of the Indian lower crust acts to drastically increase the rate of spreading near the toe of the underthrust slab and successfully correlate spatiotemporal evidence of the location of fastest slip along the Lunggar Rift in southern Tibet with rate of underthrusting. Thus, Styron et al.[6] suggest two factors drive present-day extension: a plateau-wide event driving slow extension plateau-wide since the Miocene and augmentation of that extension in southern Tibet due to the underthrust of India.

In conclusion, we argue that normal faulting and extension in Tibet results from viscous buckling of the upper crust in response to high rates of flow of a weak lower crust and does not require a resetting of the Indian boundary condition, convective removal of the mantle, nor specifically an attainment of high elevations around the Miocene. Our simulations produce a fanning of deviatoric compressional axes normal to the Himalayan front, supporting the importance of geometry of the Indian indenter, and produce high compressional deviatoric stresses along the Himalayan front as opposed to the centralized "punch" as proposed by Kapp and Guynn[8]. We interpret the patterns of dilatation and subsidence in southern and central Tibet in best-fit simulations, with a lower crustal viscosity on the order of $10^{20}$ Pa·s, as a response to the presence and high rates of flow of a weak lower crust that is limited and/or missing in the north[27–29,54,55] (Fig. 1c). Thus, in the north, the upper and lower crust deform at approximately the same rate, removing the need for viscous buckling of the upper crust and extension here is accommodated through strike-slip deformation of the whole lithosphere at lower rates (Fig. 2c).

It is important to note that our results are consistent with the TVS models that demonstrate gravitational collapse is responsible for the distributed extension within all of southern and central Tibet[1]. As the simulations move to 3-D and incorporate vertically variable strength distribution with a weaker lower crust, gravitational collapse now drives high rates of deformation and flow of the weaker lower crust that in turn induces viscous buckling of the stronger upper crust in order to deform at the same rate as the lower crust. Viscous buckling of the upper crust results in bands of east–west subsidence and dilatation that generates observed normal faulting and thinning. In each case gravitational collapse drives the observed extension in Tibet, however, in the simulations presented here, which incorporates vertical strength contrasts, extension is no longer uniformly distributed but localized within two east–west trending zones consistent with geologic observations.

## Methods

**Geodynamic simulation geometry and parameterization.** We implement a 3-D, lithospheric, finite element model (Fig. 1a) of the IN-EU collision zone in COM-SOL Multiphysics (www.comsol.com). Our spherical shell geometry encompasses major features of the IN-EU collision zone; spanning from 15 to 45° N and 68 to 110° E. The model upper surface is represented by ETOPO5 Earth topography (http://www.ngdc.noaa.gov/mgg/global/relief/ETOPO5/); the model base is

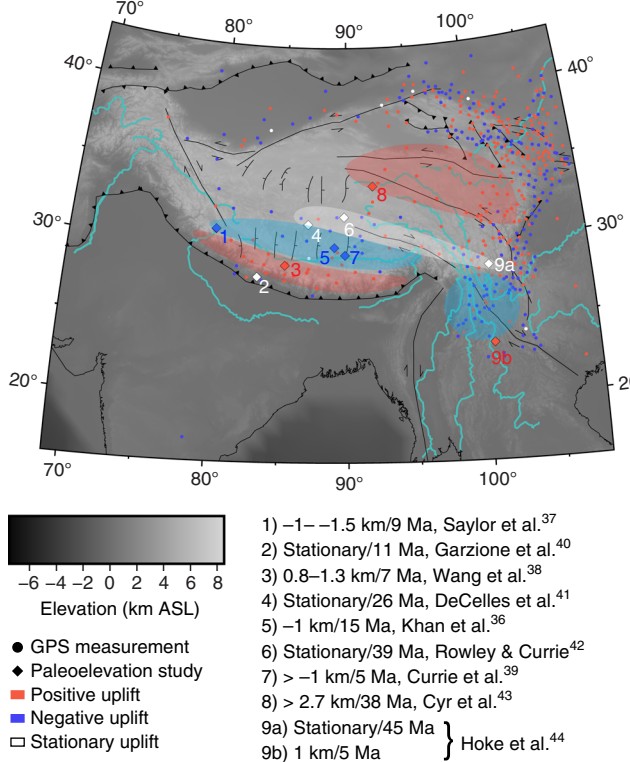

**Fig. 4** Combined evidence for positive/negative surface motion determined from GPS and paleoelevation studies. Evidence for positive (red), negative (blue), or negligible uplift across IN-EU collision zone from GPS observations (dots) and paleoelevation studies (numbered diamonds). Numbers are listed with corresponding citations and associated uplift rates. Gray-scale, black symbols, and teal lines represent topography, major faults, and rivers, respectively

Elevation (km ASL)
-6 -4 -2 0 2 4 6 8

- GPS measurement
- Paleoelevation study
- Positive uplift
- Negative uplift
- Stationary uplift

1) -1– -1.5 km/9 Ma, Saylor et al.[37]
2) Stationary/11 Ma, Garzione et al.[40]
3) 0.8–1.3 km/7 Ma, Wang et al.[38]
4) Stationary/26 Ma, DeCelles et al.[41]
5) -1 km/15 Ma, Khan et al.[36]
6) Stationary/39 Ma, Rowley & Currie[42]
7) > -1 km/5 Ma, Currie et al.[39]
8) > 2.7 km/38 Ma, Cyr et al.[43]
9a) Stationary/45 Ma
9b) 1 km/5 Ma  } Hoke et al.[44]

represented by an isoradial surface at 100 km below sea level (BSL). Three additional internal surfaces subdivide the geometry into four domains: India, Eurasia upper crust, Eurasia lower crust, and Eurasia mantle. We approximate the boundary between Indian and Eurasian plates via a subvertical surface (curved black line in Fig. 1a). To the north and west, the surface is constrained by projecting vertically down along the Himalayan Frontal Thrust (HFT) surface trace. To the east, we place the surface intersection of the Indian and Eurasian plates at the Burma Arc and constrain Burma slab dip via reported earthquake focal depths[56]. We divide the Eurasian plate into upper and lower crustal domains, with depth of the interface determined by where we place the lateral bounds for weak lower crust. For crustal regions possessing lower crustal zones of low-velocity or high conductivity, we place the upper/lower crust interface at 20 km BSL, agreeing with top of the zone of aseismicity[27] and low-velocity zones[10,57]. For all other crustal regions, we arbitrarily place the upper/lower crust divide at half the crustal thickness. We combine observations of mid-to-lower crustal low-velocity zones from receiver function analyses[49,55,58–61], joint analyses of receiver functions and Rayleigh wave dispersion[57,62,63], surface wave tomography[10,64], shear wave tomography[65], and deep seismic sounding[54] with observations of mid-to-lower crustal zones of high conductivity[9,28,66] to place the lateral bounds of weak lower crust (dashed blue line in Fig. 1c) clockwise from the HFT along the Karakorum, West Kunlun, Altyn-Tagh faults, west of the Sichuan Basin, and north of the Dien Bien Phu fault. Finally, we represent the boundary between Eurasian crust and mantle with Moho estimates of CRUST 1.0[33].

We approximate steady state, instantaneous lithospheric deformation by the equations describing Stokes-flow in an incompressible, Newtonian fluid of 3-D varying viscosity:

$$-\eta(x,y,z)\nabla^2 \mathbf{u}(x,y,z) + \nabla \mathbf{p}(x,y,z) = \mathbf{F}(x,y,z) \qquad (1)$$

$$\nabla \cdot \mathbf{u}(x,y,z) = 0 \qquad (2)$$

with dynamic viscosity of $\eta$, velocity vector $\mathbf{u}$, pressure $\mathbf{p}$, and body forces $\mathbf{F}$. 3-D varying body forces are determined by location and 3-D density variation, with densities assumed from the estimates of CRUST 1.0[33]. Vertical averages of lithospheric effective viscosity (strength) are taken from the estimates of

Flesch et al.[30] (Fig. 1c). We partition lithospheric averages between upper crust, lower crust, and mantle layers, such that the vertical integral is equal to the laterally varying lithospheric average according to the following relation:

$$\bar{\eta} = \frac{1}{L} \int_0^L \eta(r)\mathrm{d}r \qquad (3)$$

where $L$ represents lithospheric thickness (100 km + surface elevation) and $r$ integrates over all depths within the column (Fig. 1b). Assuming the ratio of upper crustal to mantle strength at any lateral point is one-to-one, and assuming test values ranging from $10^{19}$ to $10^{21}$ Pa·s for lower crustal strength, we calculate 3-D varying viscosity distributions (Fig. 1b). As a last step, we introduce zones of slab strength ($10^{22}$ Pa·s) where body wave tomography[32,67] indicates Indian and Burma slabs underthrust Tibet and Burma (dashed red line Fig. 1c). Where the zones of weak lower crust and strong slabs overlap, we suspend the base of the weak lower crust, normally the Moho, by the slab surface (e.g., Figure 1b, ~ 27° N). It is important to note because our method assumes a viscosity distribution a priori, we are unable to distinguish between power-law exponents (See Flesch et al.[1] for details) and any assumed power law will produce the same instantaneous solution for an assumed viscosity distribution.

We apply boundary conditions on the model top and bottom consistent with frictionless sliding of the lithosphere over the asthenosphere and stress-free interaction between the atmosphere and Earth's surface. We account for driving forces induced by 3-D density variations external to our model geometry by applying moving wall boundary conditions on each side wall, constrained by a continuous model velocity field determined from GPS observations and Quaternary fault slip rate data from Flesch et al.[30] variable along each side wall.

We discretize the geometry volume with mesh generation tools within COMSOL Multiphysics. Our mesh resolution varies with position; small elements correspond to places where layers thin (e.g., Southeast Asia) and domain contacts (e.g., where Indian and Eurasian plates meet). The largest mesh elements correspond to those in domain centers, where their lateral dimensions average ~ 0.5°, or about 39–54 km (depending on latitude). Mesh element shape varies, with prismatic elements used at domain boundaries transitioning to tetrahedral elements in domain interiors. We tune our mesh resolution to simultaneously correspond to our desired solution resolution and produce solutions qualitatively identical to those produced at higher resolutions. Our tuned mesh possesses 65,227 mesh elements (Fig. 1a).

**Determination of Poiseuille flow.** We develop a simple algorithm for identifying model-predicted Poiseuille flow in the lower crust. For every longitude/latitude included in the model space, we use 3-D linear interpolation to increase vertical resolution and calculate solution velocity magnitudes at all lower crustal depths. We apply a first-order, finite-impulse response (FIR), Savitzky-Golay filter to the vertical profile of solution velocity magnitudes through the lower crust; tuning filter length to yield smoothly varying profiles and minimize hot finger effects. We then sample filtered magnitudes at the top, middle, and base of the lower crust, calculating the sign of the difference from top-to-mid-layer and mid-layer-to-base magnitudes. If the difference switches sign from top-to-mid-layer and mid-layer-to-base, and the magnitude mid-layer exceeds that at the top/base, we calculate the percent of top/base magnitude in relation to mid-layer. We diagnose Poiseuille flow if both top and base percentages are less then 95% of the mid-layer magnitude. Supplementary Figure 2 illustrates examples of lower crust velocity magnitude profiles for true and false positive cases with lower crust viscosities of $10^{20}$ and $10^{19}$ Pa·s.

**Block simulations.** To validate that the source of the simulated viscous buckling in simulations with weak lower crust is not due to horizontal strength variation, we simulate two block models with simplified viscosity structure. In the first block simulation, case 1, we simulate a constant Indian lithosphere of $10^{24}$ Pa·s, Asian lithosphere of $10^{23}$ Pa·s, and weak Tibetan lower crust of $10^{20}$ Pa·s. In the second block simulation, case 2, we again simulate a constant Indian lithosphere of $10^{24}$ Pa·s, Asian lithosphere of $10^{23}$ Pa·s, and weak Tibetan lower crust of $10^{20}$ Pa·s, but then add an additional level of strength variation with Tibetan lithosphere of $10^{22}$ Pa·s. The horizontal surface velocity solutions look similar to the simulations constrained by geophysical observation based estimates of lateral strength distribution, although with an overall poorer fit to GPS observations (Table 1). The vertical surface velocity solutions are shown in Supplementary Figure 1a, b. Viscous buckling is observed in both simulations, while case 2 yields slightly higher amplitude buckling due to the weaker Tibetan lithosphere, indicating simulated viscous buckling is not due to our assumed lateral strength variation.

**Statistics.** We quantify goodness of fit for model-predicted horizontal surface velocity through the root-mean-square (RMS) and weighted-root-mean-square (WRMS) statistics. For both statistics, we use the horizontal velocity GPS

observations reported by Liang et al.[45]. We calculate RMS and WRMS misfit statistics as:

$$\mathrm{RMS_{misfit}} = \sqrt{\frac{\sum \frac{(V_m - V_o)^2}{\sigma^2}}{2N}} \qquad (4)$$

$$\mathrm{WRMS_{misfit}} = \sqrt{\frac{N}{N-1} \frac{\sum \frac{(V_m - V_o)^2}{\sigma^2}}{\sum \frac{1}{\sigma^2}}} \qquad (5)$$

respectively, where $V_m$ represents model-predicted velocity at the GPS points, $V_o$ are the GPS observations with associated uncertainty of $\sigma$, and $N$ is the number of GPS observations. For the set of GPS observations we use here[45], $N = 750$. Table 1 shows the RMS and WRMS misfits calculated for each model presented here.

## Data availability

All data used in this manuscript are freely available within the Supplementary Information sections from references provided in the paper.

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

## Acknowledgements

We benefited from discussion with Joel Saylor, Roland Burgmann, and Bill Holt. This work was supported by NSF EAR-0609337 to L.M.F.

## Author contributions

L.M.F. conceived and supervised the project and S.H.B. constructed and ran the geodynamic simulations. L.M.F. and S.H.B. wrote the paper.

## Additional information

**Competing interests:** The authors declare no competing interests.

