## [Peer Review File · Nature Communications]

Reviewers' comments:

Reviewer #1 (Remarks to the Author):

This is an interesting paper using 3D numerical modeling to demonstrate potential relationship between the enhanced ductile flow of weak lower crust and buckling of strong upper crust with normal faulting in Tibet. It is suggested on the basis of 3D modeling that the areas of normal faulting should coincide with the areas of relative subsidence resulting from upper crustal buckling. Whereas I find the numerical results important and significant, some improvements and clarifications are needed:

Novelty of this study should be better demonstrated by reviewing recent 3D numerical modeling papers concerning Himalayan collision and normal faulting (Püsök and Kaus, 2015; Chen and Gerya, 2016; Chen et al., 2017; Pang et al., 2018). In particular, Pang et al. (2018) modeled in 3D a thermomechanical relationship between the presence of weak middle/lower crust and normal faulting (which was directly modeled rather than inferred on the basis of stress estimates).

Used vertically layered viscosity structure of the lithosphere does not look realistic to me and does not agree with predominantly horizontally layered structure proposed on the basis of geodynamic inversion (Baumann and Kaus, 2015). How different result will be if only three viscosities will be used: Indian lithosphere (10^{24} Pa*s), Asian lithosphere (10^{23} Pa*s) and weak crust (10^{20} Pa*s)? Does the presence of vertical viscosity layering in the upper crust pre-defines geometry of buckling and stress orientation?

It is suggested that buckling of the upper crust is viscous. I think, this suggestion is an artifact of purely viscous P-T-stress independent rheology used in this study. Upper crustal deformation and buckling should rather be dominated by brittle/plastic rheology (e.g., Baumann and Kaus, 2015; Pang et al., 2018).

Relationship between buckling and subsidence should be investigated in more details. It could be that both are strongly related to the presence of prescribed vertical viscosity layers in the upper crust (Fig. 1b) and removing such layering will strongly change both the buckling and the stress orientation. For example, Pang et al. (2018) suggested that uplift rather than subsidence should be characteristic to the areas of normal faulting and weak middle/lower crust when uniform viscoplastic upper crust is used.

Specific comments for the paper are given below.

Taras Gerya, Zurich, 14.03.2018

Specific comments

Line 50. "Previous geodynamic simulations of lower crustal deformation in Tibet have primarily focused on estimating the viscosity required to generate observed topographic relief¹⁵ and gradients¹³ along 2D profiles, or thermomechanical derivation of viscosity with assumed flow laws and temperature gradients^{20–22}. Since 2D simulations neglect flow in and out of the third dimension and thermomechanical simulations derive strength distributions rather than test a given hypothetical distribution, neither approach addresses the lithospheric wide influence of an assumed lower crustal strength. In this work we perform 3-D lithospheric scale simulations of the India-Eurasia (IN-EU) collision zone, varying lower crustal strength for published viscosity estimates ranging from 10^{19} - 10^{21} Pa*s to explicitly determine the lithospheric surface response and assess the level of lower crustal flow, for an assumed layer-averaged lower crustal strength." Numerical modeling literature overview is not up to date. Previous research is not limited by 2D studies. There are several recent 3D numerical modeling studies concerning Tibet, including modeling of normal faulting (Püsök and Kaus, 2015; Chen and Gerya, 2016; Chen et al., 2017; Pang et al., 2018).

Fig. 1b. Vertically layered viscosity structure of the lithosphere looks unrealistic and does not agree with predominantly horizontally layered structure proposed on the basis of geodynamic inversion (Baumann and Kaus, 2015). Using purely viscous rheology for the mantle lithosphere and upper crust needs better discussion since these regions are typically assumed to be dominated

by brittle/plastic rheology (e.g., Baumann and Kaus, 2015; Chen et al., 2017; Pang et al., 2018). Indian slab geometry, which crosses Moho, also looks quite strange.

Line 88. "Additionally in simulations developing Poiseuille flow, the large strength contrast between upper and lower crust prevents deformation in the stronger upper crust from occurring at the same rate as the underlying fast-flowing, weak lower crust, causing the upper crust to viscously buckle (and references therein) to maintain plateau continuity." Viscous mechanism of buckling is a consequence of chosen simplified purely viscous P-T-stress-independent rheology of the lithosphere. Viscous buckling should not be characteristic for the upper crust which behaves predominantly brittle/plastic. Buckling should rather be visco-plastic (e.g., Baumann and Kaus, 2015; Pang et al., 2018).

Line 122. While paleoelevation estimates provide no constraints on uplift rate and age estimates of individual sites vary widely from 45 to 5 Ma, both the paleoaltimetry and geodetic results reveal an alternating pattern of margin parallel uplift and subsidence across southern Tibet roughly coincident with the surface uplift and subsidence patterns produced by simulations with weak lower crust of 1020 Pa•s and below (Figure 3c-d). "

Line 153. "Based on these correlations, we propose that normal faulting, extension and subsidence in southern and central Tibet is a result of viscous buckling of the Tibetan upper crust in response to faster rates of deformation and flow of the weak lower crust associated with gravitational collapse. Early crustal thickening and uplift of Tibet led to radioactive heating and weakening of the lower crust." Areas of normal faulting do not seem to coincide with relative topographic lows. Taken that the normal faulting was active since 13.5-4 Ma (Blisniuk et al., 2001 and references therein), the proposed relative subsidence rates of few mm/yr related to buckling (Fig. 3cii) should have produced such relative lows. In addition, Pang et al. (2018) showed that the regions of the weak middle/lower crust and normal faulting should rather be characterized by some surface uplift.

Line 138. "The locations of the gravity lows are spatially well correlated with the simulated bands of surface subsidence, corresponding to the buckling of the simulated upper crust (Figure 3ii,c-d). Thus, gravity observations may also be sensitive to the proposed buckling in the upper crust." Please explain in more detail why gravity lows may be sensitive to the buckling.

Line 144. "Origins of Normal Faulting in Southern and Central Tibet." Recently, Pang et al. (2018) modeled in 3D the relationship between normal faulting and presence of weak middle/lower crust.

References

- Baumann T., Kaus B.J.P. (2015) Geodynamic inversion to constrain the nonlinear rheology of the lithosphere. *Geophysical Journal International*. 202, 1289-1316.
- Blisniuk, P.M., Hacker, B.R., Glodny, J., Ratschbacher, L., Bi, S., Wu, Z., McWilliams, M.O., Calvert, A. (2001) Normal faulting in central Tibet since at least 13.5 Myr ago. *Nature*, 412, 628–632.
- Chen, L., Capitanio, F., Liu, L., Gerya, T. (2017) Crustal rheology controls on the Tibetan plateau formation during India-Asia convergence. *Nature Communications*, 8 Article Number: 15992.
- Chen, L., Gerya, T.V. (2016) The role of lateral lithospheric strength heterogeneities in orogenic plateau growth: Insights from 3-D thermo-mechanical modeling. *Journal of Geophysical Research*, 121, 3118-3138.
- Pang, Y., Zhang, H., Gerya, T.V., Liao, J., Cheng, H., Shi, Y. (2018) The Mechanism and Dynamics of N-S Rifting in Southern Tibet: Insight From 3-D Thermomechanical Modeling. *Journal of Geophysical Research*, 123, 859–877.
- Püsök A., Kaus B.J.P. (2015) Development of topography in 3-D continental collision models. *Geochemistry Geophysics Geosystems*. Vol. 16(5), doi: 10.1002/2015GC005732.

Reviewer #2 (Remarks to the Author):

This is a review by Lin Chen of the manuscript entitled "Normal faulting and viscous buckling in the Tibetan Plateau induced by a weak lower crust" submitted by Bischoff and Flesch for publication in Nature Communications.

This manuscript deals with an important question: what controls the development of the observed E-W extension and normal faulting in the Tibetan plateau. They used 3-D thin viscous sheet models with both horizontally and vertically variable viscosity distributions, and investigated the influence of the Tibetan lower crust viscosity on the crustal deformation and topographic expression. By comparing the predictions from the best-fit model, including the distribution of the surface velocity, strain rate and vertical motion, to observations in the Tibetan plateau, the authors argued lower crustal flow, driven by the gravitational potential energy, leads to viscous buckling of the upper crust and the observed extension in Tibet. The presented results are very interesting. I see an obvious progress relative to the traditional crustal channel flow model proposed by Royden and her colleagues twenty years ago. Therefore, I recommend publication in principle.

However, I have the following concerns which should be addressed before possible publications.

1) Recent geophysical data show that the low velocity/high electric conductivity zones are present at the mid-crustal level (20-40 km) under the Tibetan plateau (e.g., Bai et al., 2010; Yang et al., 2012; Bao et al., EPSL, 2015.). In addition, the mid-crustal felsic rocks are easier to melting than the lower crustal mafic rocks during crustal thickening (e.g., Chen & Gerya, 2016). So I was wondering how a different vertical distribution of the low-viscosity lower crust (i.e., the blue region in Fig. 1b) influences the results.

2) The authors solved steady state, instantaneous Stoke equations to model crustal deformation. No time item is involved in the modeling. The hidden logic, which link the instantaneous model-predictions to geological features, including normal faulting since Miocene and paleoelevation, should be straightforwardly described to readers.

3) In the discussion part, the authors should note that some people agree the initiation of crustal extension in Tibet occurred at the early stage of the India-Asia collision, not Miocene. They have ample evidence for this idea. See the review by Searle et al. (2011) and Wang, Q., Wyman, D.A., et al. 2010. Eocene north-south trending dikes in central Tibet: New constraints on the timing of east-west extension with implication for early plateau uplift. Earth and Planetary Science Letters, 10.1016/j.epsl.2010.07.046.

Other comments:

1) Perhaps the phrase of convective removal is better than "convective delamination".

2) Line 130, less then->less than;

3) In Figure S2, the vertical label "elevation" should be depth.

Response to Reviewers' comments:

We thank you and both the reviewers for the thorough and thoughtful reviews of the manuscript. We have revised our manuscript and have addressed all comments below.

Reviewer #1 (Remarks to the Author):

This is an interesting paper using 3D numerical modeling to demonstrate potential relationship between the enhanced ductile flow of weak lower crust and buckling of strong upper crust with normal faulting in Tibet. It is suggested on the basis of 3D modeling that the areas of normal faulting should coincide with the areas of relative subsidence resulting from upper crustal buckling. Whereas I find the numerical results important and significant, some improvements and clarifications are needed:

We thank the reviewer for his support of the manuscript and thoughtful review. We highlight the requested changes below.

Novelty of this study should be better demonstrated by reviewing recent 3D numerical modeling papers concerning Himalayan collision and normal faulting (Püsök and Kaus, 2015; Chen and Gerya, 2016; Chen et al., 2017; Pang et al., 2018). In particular, Pang et al. (2018) modeled in 3D a thermomechanical relationship between the presence of weak middle/lower crust and normal faulting (which was directly modeled rather than inferred on the basis of stress estimates).

Due to the original length requirements before transferring to *Nature Communications*, we were limited on both introduction space and number of references and only referenced the papers that directly modeled Tibet. We also note that Pang et al. (2018) was published after our initial submission. However, with the additional space we have included a more comprehensive review of previous geodynamic models and included the requested references. Please see **lines 61-64**.

Used vertically layered viscosity structure of the lithosphere does not look realistic to me and does not agree with predominantly horizontally layered structure proposed on the basis of geodynamic inversion (Baumann and Kaus, 2015).

Based on the reviews there seems to be confusion regarding the 3-D viscosity structure we are using. We step through it here to directly address the comments of the

reviewers and also added more discussion in the manuscript to clarify for all readers. See **lines 75-93**. We have also added additional labels for clarity to Figure 1.

Previous models (Flesch et al., 2001, 2018; Chen and Gerya, 2016; Pusok and Kaus, 2015; Chen et al., 2017) have demonstrated the importance of lateral strength variations geodynamic simulations in reproducing geophysical observations and features in the India/Eurasia collision zone. Therefore, we start with the vertically averaged effective viscosity field estimated from thin sheet models published in Flesch et al. (2018), which are estimated by dividing the magnitudes of the vertically averaged deviatoric stresses by the magnitudes of the kinematic strain rates (Figure 1c). The Indian plate is strong (10^{24} Pas), as are the Tarim Basin, Gobi platform, and Sichuan Basin. Overall Tibet, the Pamir, and Tien Shen are two orders of magnitude weaker than the stronger blocks (10^{22} Pas). Additionally, even weaker regions are in areas of well-developed faults (10^{21} Pas) (southern Tibet, Altyn Tagh Fault, Kunlun Fault, Chaman Fault, Xianshuihe Fault, Jiali Fault, Saigan Fault, and the Himalayan Front). These estimates of the lateral variation in effective viscosity are directly measured from both deviatoric stress estimates and GPS and geologic data. Because there is an infinite number of ways to partition strength in the vertical direction to give us back our vertical averages, we make the simple assumption that the lithospheric mantle and upper crust are of equal strength and in areas where weak lower crust has been observed either seismically or with MT observations (area within the blue dashed line Figure 1c) we assign a weaker lower crust from 18km depth to the Moho. We also impose an under thrusting of the Indian lithosphere where the Indian slab has been observed seismically (area with red dashed lines Figure 1c).

The cross section shown (note the extreme vertical exaggeration) shows the vertical strength profile at 90 degrees east. It has been shown seismically that the Indian slab is nearly absent here beneath the Himalayas as noted by the red dashed line and the small section of India here is decoupled from the Eurasia crust (Schulte-Pelkum et al., Imaging the Indian subcontinent beneath the Himalaya. **435**, 1222-1225 (2005)) thus the Indian plate is weaker here. Both seismic and MT data demonstrate the presence of a weak lower crust from the ITS to the Kunlun fault. Figure 1b shows a strong Indian plate that does not extend into Tibet and a weak lower crust extending from the Yalung-Zangpo Suture to the Kunlun. From 27-30 degrees N the Himalaya and southern Tibet are weaker and from 30-25 degrees N the Qiangtang Terrane is slightly stronger and then at 35 degrees N the model becomes weaker again at the Kunlun fault. From south to north, the Qaidam Basin is the same strength as the Qiangtang Terrane to 38 degrees N and then there is the weaker Altyn Tagh fault, and then a stronger Gobi platform and Eurasian continent. We have also labeled additional sections of the cross section in Figure 1b to help the reader identify the locations on the cross section.

It is important to note that the vertical strength partitioning as well as dynamics change along strike of the collision, therefore the cross section shown in Figure 1b is not comparable to the 2-D Bayesian models of Baumann and Kaus (2015) which is

simulated much farther west and is normal to the Himalaya front at 76 degrees E. The red and blue outlines in Figure 1c shows the seismically determined locations of both the Indian plate at depth and a weak lower crust. A cross section of our model here at 76 degrees E would look very similar to the best fit model of Baumann and Kaus. There is a significant under thrusting of the Indian plate here, and there is not evidence that the weak lower crust extends this far west. Thus, we are not using vertical layers but laterally variable vertically averaged effective viscosity and taking the next step from thin sheet modeling and incorporating 3-D structure of the Indian plate at depth and weaker lower crust in regions where both are seismically imaged. We are not imposing vertical layering but incorporating lithospheric weak zones where large lithospheric scale faults are observed. The other point to note in what appears to be differences between Figure 1b and the Bayesian models of Baumann and Kaus (2015) is the thickness of the model space each study is considering. Here we are only modeling the lithosphere and simulations extend to only 100 km depth, where as the models of Baumann and Kaus (2015) extend below 600 km depth. Additionally, in figure 14a of Baumann and Kaus (2015) there are vertical regions of weaknesses corresponding to lithospheric scale faults as well in the upper 100 km of their simulations as we include in our modeling.

*How different result will be if only three viscosities will be used: Indian lithosphere (10^{24} Pa*s), Asian lithosphere (10^{23} Pa*s) and weak crust (10^{20} Pa*s)? Does the presence of vertical viscosity layering in the upper crust pre-defines geometry of buckling and stress orientation?*

In the additional modeling we have performed that now comprises the supplementary materials, we performed two additional sets of simulations, the first as described by the reviewer above and a second set where we have the Indian lithosphere (10^{24} Pa*s), Asian lithosphere (10^{23} Pa*s) and weak crust (10^{20} Pa*s) and a stronger Tarim Basin and Gobi platform (10^{23} Pa*s) to explore models with a strong Eurasian backstop. The first important thing to note is the simplification of the viscosity field in both sets of simulations degraded the overall fit to both the predicted surface velocity field to the GPS observations as well as the crustal deviatoric stress field to observations. Again illustrating the results discussed by my authors that show the importance of lateral strength variations and pre-existing weaknesses in generating the observed deformation within the India/Eurasian collision zone. The second important result is that in both sets of simulations viscous buckling is observed at the same level and in the same areas as in the case presented within the main manuscript Figure 3. These sets of simulations reinforce our conclusions that viscous buckling occurs in the stronger upper crust when overlain by a weaker lower crust. Once the lower crust becomes weak enough to flow by Poiseuille flow the upper crust must buckle in the areas of rapidly deforming lower crust in order to maintain plateau continuity. Thus the length scale and lateral extent of the buckling is related to the region of weak lower crust. This discussion has been strengthened in the manuscript **see lines 177-196**.

It is suggested that buckling of the upper crust is viscous. I think, this suggestion is an artifact of purely viscous P-T-stress independent rheology used in this study. Upper crustal deformation and buckling should rather be dominated by brittle/plastic rheology (e.g., Baumann and Kaus, 2015; Pang et al., 2018).

We observe buckling as a consequence of the instability generated by the vertical stress associated with gravitational potential energy variations resulting from the development of topography and thickening of the Tibetan crust and viscosity contrasts between the upper and lower crust. This buckling happens in the context of a viscous simulation used to estimate the accumulation of stress and deformation accumulated over multiple seismic cycles and so we are unable to comment on other rheologies.

Relationship between buckling and subsidence should be investigated in more details. It could be that both are strongly related to the presence of prescribed vertical viscosity layers in the upper crust (Fig. 1b) and removing such layering will strongly change both the buckling and the stress orientation. For example, Pang et al. (2018) suggested that uplift rather than subsidence should be characteristic to the areas of normal faulting and weak middle/lower crust when uniform visco-plastic upper crust is used.

We find that the buckling is related to presence of a weak lower crust and the strength contrast between the upper and lower crust not the assumed viscosity field as discussed above and highlighted in the additional simulations. This result is similar to that by Pang et al. (2018) that thickening, uplift and rifting occurred only in regions with a weak lower crust. Both studies highlighted the need for a weaker lower crust in generating rifting and extension. In many ways the paper of Pang et al and our manuscript are very complementary and in others it is difficult to make a direct comparison due to the difference in length scale both studies. Here, our simulations are continent scale whereas Pang et al. model only southern Tibet. Here extension and flow of the lower crust is driven by intrinsic density variations within the lithosphere that generate gravitational collapse and drive lower crustal flow. Pang et al. apply an E-W tensional stress field boundary condition as a proxy for gravitational collapse.

Here we have a weak lower crust over all of the proper Tibetan plateau as is observed with seismic and MT data. Pang et al have a weak lower crust that only extends 150 and 260 km in the N-S direction depending on the model simulation. Because of the differences in scale and extent of the weaker lower crust in each study, each study provides unique insights into the origin of normal faulting in southern Tibet through different proposed hypothesis. Each study is valid and they are difficult to compare side to side. We discuss the results of the Pang study and try to put it into context with our own results. **See line 266.**

Specific comments for the paper are given below.

Taras Gerya, Zurich, 14.03.2018

Specific comments

Line 50. "Previous geodynamic simulations of lower crustal deformation in Tibet have primarily focused on estimating the viscosity required to generate observed topographic relief¹⁵ and gradients¹³ along 2D profiles, or thermomechanical derivation of viscosity with assumed flow laws and temperature gradients^{20–22}. Since 2D simulations neglect flow in and out of the third dimension and thermomechanical simulations derive strength distributions rather than test a given hypothetical distribution, neither approach addresses the lithospheric wide influence of an assumed lower crustal strength. In this work we perform 3-D lithospheric scale simulations of the India-Eurasia (IN-EU) collision zone, varying lower crustal strength for published viscosity estimates ranging from 10^{19} - 10^{21} Pa·s to explicitly determine the lithospheric surface response and assess the level of lower crustal flow, for an assumed layer-averaged lower crustal strength."

Numerical modeling literature

overview is not up to date. Previous research is not limited by 2D studies. There are several recent 3D numerical modeling studies concerning Tibet, including modeling of normal faulting (Püsök and Kaus, 2015; Chen and Gerya, 2016; Chen et al., 2017; Pang et al., 2018).

We have included a wider discussion of the previous models in the literature with the extra space. See lines 62-64.

Fig. 1b. Vertically layered viscosity structure of the lithosphere looks unrealistic and does not agree with predominantly horizontally layered structure proposed on the basis of geodynamic inversion (Baumann and Kaus, 2015). Using purely viscous rheology for the mantle lithosphere and upper crust needs better discussion since these regions are typically assumed to be dominated by brittle/plastic rheology (e.g., Baumann and Kaus, 2015; Chen et al., 2017; Pang et al., 2018). Indian slab geometry, which crosses Moho, also looks quite strange.

Please see detailed discussion above on 3-D viscosity distribution to the reviewers general point. See lines 75-93.

Line 88. "Additionally in simulations developing Poiseuille flow, the large strength contrast between upper and lower crust prevents deformation in the stronger upper crust from occurring at the same rate as the underlying fast-flowing, weak lower crust, causing the upper crust to viscously buckle (and references therein) to maintain plateau continuity." Viscous mechanism of buckling is a consequence of chosen simplified purely viscous P-T-stress-independent rheology of the lithosphere. Viscous buckling should not be characteristic for the upper crust which behaves predominantly

brittle/plastic. Buckling should rather be visco-plastic (e.g., Baumann and Kaus, 2015; Pang et al, 2018).

We observe buckling as a consequence of the instability generated by the vertical stress associated with gravitational potential energy variations resulting from the development of topography and thickening of the Tibetan crust and viscosity contrasts between the upper and lower crust. This buckling happens in the context of a viscous simulation used and estimates the accumulation of stress and deformation accumulated over multiple seismic cycles and so we are unable to comment on other rheologies.

Line 122. While paleoelevation estimates provide no constraints on uplift rate and age estimates of individual sites vary widely from 45 to 5 Ma, both the paleoaltimetry and geodetic results reveal an alternating pattern of margin parallel uplift and subsidence across southern Tibet roughly coincident with the surface uplift and subsidence patterns produced by simulations with weak lower crust of 10²⁰ Pa•s and below (Figure 3c-d). “

Line 153. “Based on these correlations, we propose that normal faulting, extension and subsidence in southern and central Tibet is a result of viscous buckling of the Tibetan upper crust in response to faster rates of deformation and flow of the weak lower crust associated with gravitational collapse. Early crustal thickening and uplift⁴⁸ of Tibet led to radioactive heating and weakening of the lower crust¹⁹.”

Areas of normal faulting do not seem to coincide with relative topographic lows. Taken that the normal faulting was active since 13.5-4 Ma (Blisniuk et al., 2001 and references therein), the proposed relative subsidence rates of few mm/yr related to buckling (Fig. 3cii) should have produced such relative lows. In addition, Pang et al. (2018) showed that the regions of the weak middle/lower crust and normal faulting should rather be characterized by some surface uplift.

The review is correct that areas of observed normal faulting does not correlate with topographic lows, however in southern Tibet they do correlate with geologic data that shows the entire region of southern Tibet has subsided over the past 10-5 Ma and is bounded by areas of geologic uplift. A similar pattern is observed in the best estimates of vertical GPS. These areas also correlate with predicted dilatational stresses that will generate normal faults and subsidence due to buckling. Unfortunately, the second set of normal faulting observed in a region where both geologic investigation and geophysical deployments are difficult and there is no vertical rate deformation here to compare with our models. There is however observed normal faulting and where our model predicts dilatation in response to the buckling. The reviewer is correct that a few mm/yr would result in large topographic changes, which is our main argument for the exclusion of a lower crustal viscosity lower than 10²⁰ that produces vertical rates in excess of observed. Color shading in Figure 3cii shows subsidence rates ranging from 0.5-2 mm/yr. Overall vertical viscosity rates of 0.5 mm/yr to 1 mm/yr are predicted by

the paleoelevation estimates for our preferred model with lower crustal viscosity of 10^{20} Pa*s. For the simulations of 10^{21} Pa* vertical velocity rates are ~ 0 . Therefore, we expressed the preferred model with a lower crustal velocity on the order of 10^{20} Pa*s in actuality the lower crust is bounded by a the lowest value of 10^{20} Pa*s and not as high as 21 Pa*s. See **line 20**. See original text **lines 210-219**. We add Pang et al. (2018) to discussion the discussion, **line 266**.

Line 138. "The locations of the gravity lows are spatially well correlated with the simulated bands of surface subsidence, corresponding to the buckling of the simulated upper crust (Figure 3ii,c-d). Thus, gravity observations may also be sensitive to the proposed buckling in the upper crust." Please explain in more detail why gravity lows may be sensitive to the buckling.

We further add to the discussion on correlation with the spatial scale of gravity lows that have been interpreted to be buckling of the lithospheric mantle with the spatial scale of our predicted upper crustal buckling. If the crust and the mantle are of the same strength as they are in the simulations presented here, when Poiseuille flow of the lower crust develops the lithospheric mantle would likewise buckle for plateau continuity and because the spatial scale is controlled by the present of weak lower crust one would expect them to correlate. This has been added to the discussion **see lines 230-234**.

Line 144. "Origins of Normal Faulting in Southern and Central Tibet." Recently, Pang et al. (2018) modeled in 3D the relationship between normal faulting and presence of weak middle/lower crust.

We have added the results of Pang et al. to the discussion, **see line 266**. Again, this manuscript had not been published at the time of our original submission.

References

- Baumann T., Kaus B.J.P. (2015) Geodynamic inversion to constrain the nonlinear rheology of the lithosphere. *Geophysical Journal International*. 202, 1289-1316.
- Blisniuk, P.M., Hacker, B.R., Glodny, J., Ratschbacher, L., Bi, S., Wu, Z., McWilliams, M.O., Calvert, A. (2001) Normal faulting in central Tibet since at least 13.5 Myr ago. *Nature*, 412, 628–632.
- Chen, L., Capitanio, F., Liu, L., Gerya, T. (2017) Crustal rheology controls on the Tibetan plateau formation during India-Asia convergence. *Nature Communications*, 8 Article Number: 15992.
- Chen, L., Gerya, T.V. (2016) The role of lateral lithospheric strength heterogeneities in orogenic plateau growth: Insights from 3-D thermo-mechanical modeling. *Journal of Geophysical Research*, 121, 3118-3138.
- Pang, Y., Zhang, H., Gerya, T.V., Liao, J., Cheng, H., Shi, Y. (2018) The Mechanism

and Dynamics of N-S Rifting in Southern Tibet: Insight From 3-D Thermomechanical Modeling. *Journal of Geophysical Research*, 123, 859–877.
Püsök A., Kaus B.J.P. (2015) Development of topography in 3-D continental collision models. *Geochemistry Geophysics Geosystems*. Vol. 16(5), doi:10.1002/2015GC005732.

Reviewer #2 (Remarks to the Author):

This is a review by Lin Chen of the manuscript entitled "Normal faulting and viscous buckling in the Tibetan Plateau induced by a weak lower crust" submitted by Bischoff and Flesch for publication in Nature Communications.

This manuscript deals with an important question: what controls the development of the observed E-W extension and normal faulting in the Tibetan plateau. They used 3-D thin viscous sheet models with both horizontally and vertically variable viscosity distributions, and investigated the influence of the Tibetan lower crust viscosity on the crustal deformation and topographic expression. By comparing the predictions from the best-fit model, including the distribution of the surface velocity, strain rate and vertical motion, to observations in the Tibetan plateau, the authors argued lower crustal flow, driven by the gravitational potential energy, leads to viscous buckling of the upper crust and the observed extension in Tibet. The presented results are very interesting. I see an obvious progress relative to the traditional crustal channel flow model proposed by Royden and her colleagues twenty years ago. Therefore, I recommend publication in principle. However, I have the following concerns which should be addressed before possible publications.

We thank the reviewer for his constructive and thoughtful review and address each specific concern raised below.

1) Recent geophysical data show that the low velocity/high electric conductivity zones are present at the mid-crustal level (20-40 km) under the Tibetan plateau (e.g., Bai et al., 2010; Yang et al., 2012; Bao et al., EPSL, 2015.). In addition, the mid-crustal felsic rocks are easier to melting than the lower crustal mafic rocks during crustal thickening (e.g., Chen & Gerya, 2016). So I was wondering how a different vertical distribution of the low-viscosity lower crust (i.e., the blue region in Fig. 1b) influences the results.

As we note above, viscous buckling occurs in region of upper crust overlain by a weaker lower crust so the lateral extent is controlled by where there is weaker lower crust.

It is the development of Poiseuille flow and the differential rates of deformation that induce viscous buckling and flow of the lower crust driven by gravitational collapse of the high plateau. Therefore, there would be a trade-off between layer thickness of a weaker lower crust and strength of the lower crust in generating buckling. A thicker weak lower crust would not be able to support the high plateau topography as a thin lower crustal layer of the same strength, thus a thicker layer would form more wide-scale regions where the lower crust is deforming under Poiseuille flow.

2) The authors solved steady state, instantaneous Stoke equations to model crustal deformation. No time item is involved in the modeling. The hidden logic, which link the instantaneous model-predictions to geological features, including normal faulting since Miocene and paleoelevation, should be straightforwardly described to readers.

The reviewer is correct that we use an instantaneous viscous rheology to estimate the accumulation of stress and deformation accumulated over multiple seismic cycles. We have added the discussion on this link in **lines 75-77**.

3) In the discussion part, the authors should note that some people agree the initiation of crustal extension in Tibet occurred at the early stage of the India-Asia collision, not Miocene. They have ample evidence for this idea. See the review by Searle et al. (2011) and Wang, Q., Wyman, D.A., et al. 2010. Eocene north–south trending dikes in central Tibet: New constraints on the timing of east–west extension with implication for early plateau uplift. Earth and Planetary Science Letters, 10.1016/j.epsl.2010.07.046.

We have added discussion on the timing of normal faulting in Tibet **see lines 272-273**.

Other comments:

1) Perhaps the phrase of convective removal is better than "convective delamination". We have made the requested change.

*2) Line 130, less then->less than;
Done.*

*3) In Figure S2, the vertical label "elevation" should be depth.
Done.*

Reviewers' comments:

Reviewer #1 (Remarks to the Author):

The Authors did a careful work for taking into account reviewers' comment and the paper is now ready for publication.

Taras Gerya, Zurich, 03.07.2018

Reviewer #2 (Remarks to the Author):

I looked at the revision and read the paper with interest once again. The quality of the manuscript has been improved considerably after revision, in particular by running two additional models. However, one of my concerns is not addressed directly. That is how a different vertical distribution of the low-viscosity layer influences viscous buckling of the upper crust. Since the authors aim at modeling the crustal deformation in the Tibetan plateau, some well-constrained observations should be taken into account. For example, large portions of the Tibetan plateau are underlain by rocks that have upper crustal compositions and physical properties (e.g., Owens and Zandt, 1997; Hacker et al., 2000; DeCelles et al., 2002). Both recent petrological and geophysical data show that it is the middle crust (at depths of 20-40 km), rather than the lower crust, beneath the plateau is partially molten (Wang et al., 2012, *Journal of Petrology*, 53: 2523-2566; Yang et al., 2012; Bai et al., 2010). If a model with a weak middle crust layer predicts similar results as the best-fit model, the conclusion will be more convincing. At the moment, this type of testing is missing.

The other point is related to the discussion. The authors demonstrate that a weak lower crust with the viscosity of 10^{20} Pa s is a pre-requisite for reproducing the surface motion and crustal deformation in the present-day Tibetan plateau. But what such a weak lower crust corresponds to in nature is not mentioned. In addition, if the vertical velocities shown in Figure 3cii have worked since the late Miocene, the southern Tibet would have a large undulation. But this is not the case. Perhaps gravitational collapse helps smooth the surface. Anyway, I think an in-depth discussion of the relevance of the models on the geological constraints is needed.

Other comments:

1) Line 79, 'Tien Shen' -> Tien Shan.

2) Line 126-143, these are new model results, but indications to new Figure(s) is missing.

3) In the supplementary material, what are 'moving wall conditions'? Please explain in more detail what the lateral boundary conditions are.

Lin Chen

Reviewer #1 (Remarks to the Author):

The Authors did a careful work for taking into account reviewers' comment and the paper is now ready for publication.

Taras Gerya, Zurich, 03.07.2018

We thank the reviewer for his time in helping to improve the quality of the manuscript.

Reviewer #2 (Remarks to the Author):

I looked at the revision and read the paper with interest once again. The quality of the manuscript has been improved considerably after revision, in particular by running two additional models.

We thank the reviewer for his thoughtful comments and careful review.

However, one of my concerns is not addressed directly. That is how a different vertical distribution of the low-viscosity layer influences viscous buckling of the upper crust. Since the authors aim at modeling the crustal deformation in the Tibetan plateau, some well-constrained observations should be taken into account. For example, large portions of the Tibetan plateau are underlain by rocks that have upper crustal compositions and physical properties (e.g.,

Owens and Zandt, 1997; Hacker et al., 2000; DeCelles et al., 2002). Both recent petrological and geophysical data show that it is the middle crust (at depths of 20-40 km), rather than the lower crust, beneath the plateau is partially molten (Wang et al., 2012, Journal of Petrology, 53: 2523-2566; Yang et al., 2012; Bai et al., 2010). If a model with a weak middle crust layer predicts similar results as the best-fit model, the conclusion will be more convincing. At the moment, this type of testing is missing.

We agree with the reviewer that a model demonstrating the effect of decreasing the weak lower crustal layer that is able to generate viscous buckling is important to demonstrate in order for the conclusion to be more convincing. We studied the manuscripts listed above and find that for everywhere in Tibet and surroundings regions the weak lower crust extends to 50 km, with the exception of the Changtang terrane where the thickness is ~45km (Yang et al., 2012). Other studies show the thickness of the weaker lower layer extending to or beyond 50 km depth everywhere we have lower weak lower crust in our model (Yao et al., 2010; Bai et al., 2010; Huang et al., 2010; Jiang et al., 2014). Therefore, for an additional model we kept the layer of the weak lower crust constant extending from 20km-50km depth. Results from this model are shown in Figure S3c. It is important to note that using a constant layer of weak lower crustal thickness again produces the viscous buckling. Providing further evidence that viscous buckling is dependent on having a weak layer and thinning that layer to the narrowest estimates from geophysical data still generates viscous buckling. This discussion has been added to the main text, see **lines 144-153**.

The other point is related to the discussion. The authors demonstrate that a weak lower crust with the viscosity of 10^{20} Pa s is a pre-requisite for reproducing the surface motion and crustal deformation in the present-day Tibetan plateau. But what such a weak lower crust corresponds to in nature is not mentioned.

We believe that placing bounds on the viscosity of the lower crust does add to the discussion on what is happening in nature. In the abstract and introduction several models of Tibet argue for a wide range of lower crustal viscosities over 6 orders of magnitude. There are several review papers (See Klemperer 2006) that argue for significant movement of the lower crust. Simply placing bounds on the viscosity of the lower crust and the levels of lower crustal flow as in Figure 3 and as discussed in the manuscript has significant implications as to the allowable processes within the Tibetan lithosphere. Additionally, starting with a lower crustal viscosity of 10^{21} Pas and assuming other parameters then expanding models out petrologically to say something geologic about the lower crust is beyond the scope of this manuscript.

In addition, if the vertical velocities shown in Figure 3cii have worked since the late Miocene, the southern Tibet would have a large undulation. But this is not the case. Perhaps gravitational collapse helps smooth the surface. Anyway, I think an in-depth discussion of the relevance of the models on the geological constraints is needed.

This was an issue also raised by Reviewer #1 in the first round of revisions. The reviewer is correct that if the rates shown in Figure 3cii if exact Tibet would have a larger undulation then is

presently observed. However, undulations are observed as illustrated by geologic data that shows the entire region of southern Tibet has subsided over the past 10-5 Ma and is bounded by areas of geologic uplift. A similar pattern is observed in the best estimates of vertical GPS. These areas also correlate with predicted dilatational stresses that will generate normal faults and subsidence due to buckling. Unfortunately, the second set of normal faulting observed in a region where both geologic investigation and geophysical deployments are difficult and there is no vertical rate deformation here to compare with our models. Overall vertical velocity rates of 0.5 mm/yr to 1 mm/yr are predicted by the paleoelevation estimates for our preferred model with lower crustal viscosity of 10^{20} Pa*s, larger than what is observed. For the simulations of 10^{21} Pa* vertical velocity rates are ~ 0 and produce no undulations. Therefore, we expressed the preferred model with a lower crustal velocity on the order of 10^{20} Pa*s in actuality the lower crust is bounded by the lowest value of 10^{20} Pa*s and not as high as 21 Pa*s. See original text **lines 20-21; 193; 241**.

Other comments:

1) Line 79, 'Tien Shen' -> Tien Shan.

Corrected.

2) Line 126-143, these are new model results, but indications to new Figure(s) is missing.

Figure references have been added.

3) In the supplementary material, what are 'moving wall conditions'? Please explain in more detail what the lateral boundary conditions are.

We have added detail.

Lin Chen

REVIEWERS' COMMENTS:

Reviewer #2 (Remarks to the Author):

I am satisfied with the Authors' revision, and recommend publication as it is.

REVIEWERS' COMMENTS:

Reviewer #2 (Remarks to the Author):

I am satisfied with the Authors' revision, and recommend publication as it is.

We thank the reviewer for the time and effort to review and improve the manuscript.